# Neural Matching Fields: Implicit Representation of Matching Fields for Visual Correspondence

**Sunghwan Hong**
Korea University

**Jisu Nam**
Korea University

**Seokju Cho**
Korea University

**Susung Hong**
Korea University

**Sangryul Jeon**
UC Berkeley

**Dongbo Min**
Ewha Womans University

**Seungryong Kim**[*]
Korea University

## Abstract

Existing pipelines of semantic correspondence commonly include extracting high-level semantic features for the invariance against intra-class variations and background clutters. This architecture, however, inevitably results in a low-resolution matching field that additionally requires an ad-hoc interpolation process as a post-processing for converting it into a high-resolution one, certainly limiting the overall performance of matching results. To overcome this, inspired by recent success of implicit neural representation, we present a novel method for semantic correspondence, called Neural Matching Field (NeMF). However, complicacy and high-dimensionality of a 4D matching field are the major hindrances, which we propose a cost embedding network to process a coarse cost volume to use as a guidance for establishing high-precision matching field through the following fully-connected network. Nevertheless, learning a high-dimensional matching field remains challenging mainly due to computational complexity, since a naïve exhaustive inference would require querying from all pixels in the 4D space to infer pixel-wise correspondences. To overcome this, we propose adequate training and inference procedures, which in the training phase, we randomly sample matching candidates and in the inference phase, we iteratively performs PatchMatch-based inference and coordinate optimization at test time. With these combined, competitive results are attained on several standard benchmarks for semantic correspondence. Code and pre-trained weights are available at `https://ku-cvlab.github.io/NeMF/`.

## 1 Introduction

Establishing visual correspondence across semantically similar images is a fundamental problem in computer vision, which has been facilitating many applications including visual localization [68, 38], structure-from-motion [69], image editing [1] and autonomous driving [33]. Unlike traditional dense correspondence tasks [20, 23], where visually similar images of the same scene are used as inputs, semantic correspondence problem poses additional challenges due to intra-class appearance and severe geometry variations among object instances [15, 16].

Much research [65, 48, 29, 25, 32, 50, 64, 26, 47, 31, 8, 35, 77, 9] in semantic correspondence literature attempt to address above challenges by leveraging Convolutional Neural Networks (CNNs)-based features thanks to their greater semantic invariance than traditional hand-crafted descriptors [37, 10, 66, 3] that only capture low-level local structure. As shown in Fig. 1(a), they typically perform matching with deeper features that contain high-level semantics to obtain a low-resolution correspondence map, and they are enforced to use hand-crafted interpolation techniques, *e.g.,* bilinear

---

[*]Corresponding author

36th Conference on Neural Information Processing Systems (NeurIPS 2022).

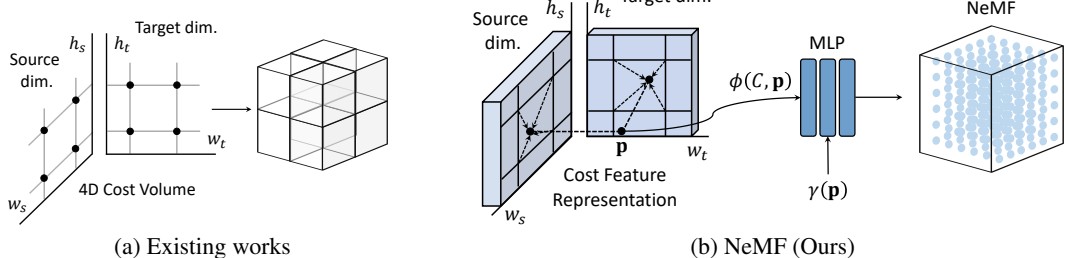

(a) Existing works            (b) NeMF (Ours)

Figure 1: **Intuition of the proposed neural matching field (NeMF):** (a) existing works [47, 8] and (b) the proposed NeMF. Unlike existing methods that explicitly compute and store discrete matching field defined at low resolution, we implicitly represent a high-dimensional 4D matching field with deep fully-connected networks defined at arbitrary original image resolution.

interpolation [65, 64, 26, 8, 77, 9] or TPS warping with sparse keypoints [48, 50, 47], significantly reducing localization precision in matching details. Instead of these hand-crafted designs, several works [25, 76, 19] attempted to formulate a coarse-to-fine approach by utilizing multi-level features, but they often suffer from the propagation of initial error from the early coarse level.

Inspired by recent success of Implicit Neural Representation (INR) [44, 57, 73, 45, 5, 51] where a coordinate-based neural network allow to model a continuous field, we propose a novel learnable framework, dubbed Neural Matching Field (NeMF), that aims to establish high-precision correspondence at arbitrary original image resolution. However, typically, matching field between a pair of images is complicated and high-dimensional, where a simple fully-connected network, which is commonly used in INR, may fail to implicitly represent such a high-dimensional matching field. To better structure the intricate matching field, we propose a cost embedding network that takes a coarse cost volume to learn cost feature representation and use it as a guidance for generating high-precision matching field through the following fully-connected network. This is accomplished by designing the cost embedding network with convolutions [18] and self-attention layers [78] to encapsulate local contexts and impart to all pixels with global receptive fields of self-attention, which also helps to compensate for the lack of inductive bias of Transformer by injecting convolutional inductive bias. The intuition of the proposed method is illustrated in Fig. 1(b).

Although leveraging a cost representation may alleviate the issues for learning the matching field with details preserved, naïvely performing feed-forward for all pixels of matching field to find pixel-wise correspondences that are used for providing supervisory signals or inference would be computationally intractable. To this end, we learn a neural matching field by enforcing the network to predict the correctness of a correspondence given a set consisting of randomly sampled points and the ground-truth point. Furthermore, as the intractability issue applies similarly at inference phase, we propose a novel test-time optimization method that not only adopts a PatchMatch [1]-based search space sampling strategy in the learned neural matching, but also optimizes the coordinates for a means of correction that lead to find better correspondences as the iteration progresses. We alternatively perform both PatchMatch-like inference and coordinate optimization, which works as an exploration and exploitation solution.

In the experiments, we evaluate the effectiveness of the proposed method using the standard benchmarks for semantic correspondence [49, 15, 16]. We demonstrate that the proposed implicit matching field effectively finds high precision correspondences, reporting the dramatically boosted performances in comparison to that of hand-crafted interpolation techniques. We also conduct extensive ablation study to validate our design choices and explore the effectiveness of each components.

## 2   Related Work

**Semantic Correspondence.**    The earliest works [10, 66, 3, 37] focused on feature extraction stage by proposing the hand-crafted feature descriptors. Although these works are probably based on the most well-known traditional hand-crafted feature descriptors, they exhibit limited capability to capture high-level semantics. Resolving such an issue, convolutional neural networks (CNNs) [72, 18] made a paradigm shift thanks to their robust representations to deformations, at first replacing the hand-crafted features with deep features, rapidly converging towards end-to-end learning. Since

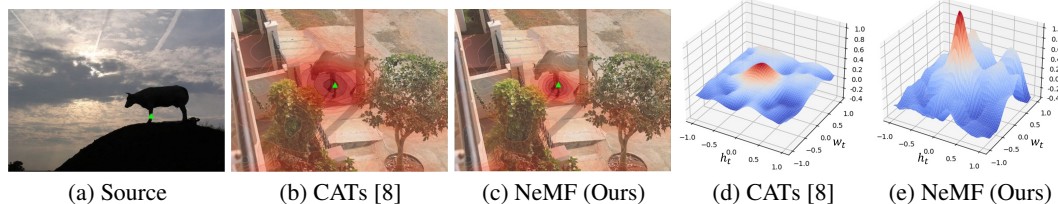

|(a) Source|(b) CATs [8]|(c) NeMF (Ours)|(d) CATs [8]|(e) NeMF (Ours)|

Figure 2: **Visualization of matching fields:** (a) source image, where the keypoint is marked as green triangle, (b), (c) 2D contour plots of cost by CATs [8] and the NeMF (ours), respectively, and (d), (e) 3D visualization of cost by CATs [8] and NeMF, with respect to the keypoint in (a). Note that all the visualizations are smoothed by a Gaussian kernel. Compared to CATs [8], NeMF has higher peak near ground-truth and makes a more accurate prediction.

then, leveraging deep features has become the *de facto* standard. Rocco *et al.* [62] first proposed an end-to-end geometric matching networks based on the deep feature maps extracted using CNNs and correlation maps computed between the extracted feature maps. Since then, using correlation map which contains all pairs of similarities between descriptors has become popular by numerous matching networks [65, 63, 43, 76, 19, 31, 48, 50, 47, 64, 26, 34, 82, 32, 40, 8]. However, not only exhaustively computing and storing all pairwise similarities require quadratic memory and computation complexity with respect to the input spatial size, which is a major downside, but also it is infeasible to compute them as the resolution increases. This inevitably caused existing methods to utilize correlation map defined at low resolutions.

On the other hand, notable methods include NC-Net [65] which first proposes to employ 4D convolutions to identify spatially consistent matches by exploring neighbourhood consensus. DHPF [50] applied probabilistic Hough matching (PHM) [7] to find the matching points. CHM [47] extends the PHM by employing high-dimensional convolutional kernels to aggregate 6D correlation maps. CATs [8] and its extension [9] use transformers [78, 12] to explore global consensus from correlation maps thanks to transformers' ability to consider long-range interactions. All these works exploit rich semantics present at high-level features for robust matching across semantically similar images. However, this inevitably necessitates up-sampling the predicted correspondence field to original image-level resolutions, which may result in losing precision. Unlike them, we implicitly represent a matching field at arbitrary image-level resolution, eliminating such a loss and ensure high-precision correspondence field to be found, as shown in Fig. 2.

**Implicit Neural Representation.** Implicit neural representation (INR), also known as coordinate-based representations, is continuous, differentiable signal representation parameterized by neural networks [45]. INR recently received huge attention, and substantial progress has been made in this direction. INR is not coupled to spatial resolution, making the memory requirements to parameterize the input signals orthogonal to spatial resolution.

Notable contributions to INR include COIN [13] that proposes a compression method with INR and LIIF [5] learns a continuous representation for images that can be presented in arbitrary resolution. DeepSDF [53] was a pioneering work that enables high quality representation from 3D input data by leveraging a learned continuous signed distance function. Occupancy networks [44] implicitly represent the 3D surface as the continuous decision boundary. IM-Net [6] also takes a similar approach, learning a mapping from coordinates conditioned by shape feature vectors to determine whether a point is outside or inside the 3D shape.

Since then, INR-based works consistently have attained state-of-the-art performance in 3D computer vision. As a pioneering work, NeRF [45] represents 3D scenes as neural radiance fields for novel view synthesis. Inspired by NeRF, a large number of works [81, 54, 70, 28, 55, 59, 14, 79, 42, 58, 2, 73, 4, 51, 80, 61, 5] made a progress in this direction. Although flourished in 3D computer vision tasks, INR has never been properly studied or explored in visual correspondence tasks, which we address in this work.

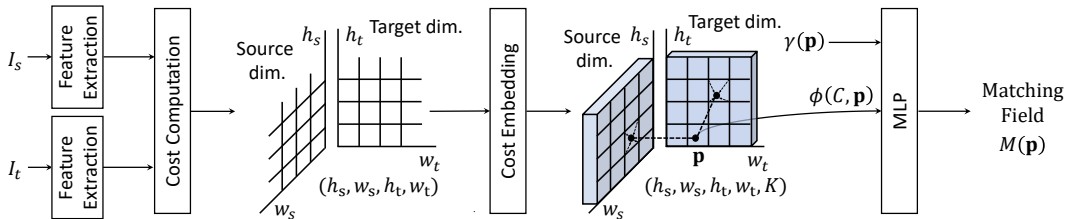

Figure 3: **Overview of network architecture:** Given a pair of images as an input, we first extract features using CNNs [18] and compute an initial noisy cost volume at low resolution. We feed the noisy cost volume with the proposed encoder consisting of convolution [18] and Transformer [78], and decode with deep fully connected networks by taking the encoded cost and coordinates as inputs.

## 3 Preliminary

Neural Radiance Field (NeRF) is a continuous function $f_\omega$ with parameters $\omega$ that computes a volume density $\sigma$ and RGB color value $\mathbf{c} = (r, g, b)$ taking as an input a 3D location $\mathbf{o} = (x, y, z)$ and 2D viewing direction $\mathbf{d} = (\theta, \phi)$, such that $f_\omega : (\mathbf{o}, \mathbf{d}) \rightarrow (\sigma, \mathbf{c})$. In practice, as shown in [45, 74], mapping low dimensional inputs $\mathbf{x}$ and $\mathbf{d}$ to higher dimensional features before passing them through the neural networks enables better representing high frequency variations.

Specifically, denoting $\gamma(\cdot)$ as an encoding function and $L$ as the number of frequency octaves as

$$\gamma(t) = [\sin(2^0 t\pi), \cos(2^0 t\pi), ..., \sin(2^L t\pi), \cos(2^L t\pi)], \tag{1}$$

the overall process $f_\omega : \mathbb{R}^{L_\mathbf{o}} \times \mathbb{R}^{L_\mathbf{d}} \rightarrow \mathbb{R}^+ \times \mathbb{R}^3$ is defined as

$$\{\sigma, \mathbf{c}\} = f_\omega(\gamma(\mathbf{o}), \gamma(\mathbf{d})) \tag{2}$$

where $L_\mathbf{o}$ and $L_\mathbf{d}$ denote output dimension of the encoded coordinate $\mathbf{o}$ and viewing direction $\mathbf{d}$, respectively. The function $f_\omega$ is formulated as a fully-connected deep network. This implicit neural representations are not coupled to spatial resolution for using continuous functions, making the memory consumption required to parameterize the signal independent of spatial resolution [45, 74].

## 4 Neural Matching Fields (NeMF)

### 4.1 Problem Statement and Overview

The overview of NeMF is shown in Fig. 3. Given a pair of source and target images as $I_s \in \mathbb{R}^{H_s \times W_s}$ and $I_t \in \mathbb{R}^{H_t \times W_t}$, our objective is to find a dense correspondence field $F(\mathbf{x})$ that is defined for each pixel $\mathbf{x}$ in original image resolution, which warps $I_s$ towards $I_t$ so as to satisfy $I_t(\mathbf{x}) \approx I_s(\mathbf{x} + F(\mathbf{x}))$.

Following traditional matching pipeline, we first extract dense features $D_s$ and $D_t$ from the input source and target images, and then compute full pair-wise similarity scores between them using cosine distance such that:

$$C(\mathbf{x}, \mathbf{y}) = \frac{D_s(\mathbf{x}) \cdot D_t(\mathbf{y})}{\|D_s(\mathbf{x})\|\|D_t(\mathbf{y})\|}, \tag{3}$$

where $\mathbf{x} \in [0, h_s) \times [0, w_s)$ and $\mathbf{y} \in [0, h_t) \times [0, w_t)$, and $\|\cdot\|$ denotes $l$-2 normalization. Previous approaches [50, 47, 8] extracted features from the deep layers of CNN to have high-level semantic invariance, resulting the spatial resolution of $D_s$ and $D_t$ to be reduced, i.e., $h < H$ and $w < W$. Consequently, utilizing the coarse similarity scores $C(\mathbf{x}, \mathbf{y})$ to infer correspondences inevitably yields a low-resolution correspondence map, which additionally requires post-processing to be interpolated into a high-resolution map [8, 47].

To alleviate this issue, we propose an INR-based learnable framework, called neural matching field (NeMF), that implicitly represents a high-dimensional 4D matching field to infer high-precision correspondences at arbitrary scales without any post-processing procedure. Specifically, we formulate a continuous function $f_\theta$ as a multi layer perceptron (MLP) with parameters $\theta$ in which encoded position and its corresponding cost feature vector are taken as an input. Formally, denoting 4D coordinates defined in original image resolution as $\mathbf{p} = [\mathbf{x}, \mathbf{y}]$ where $\mathbf{x} \in [0, H_s) \times [0, W_s)$ and $\mathbf{y} \in [0, H_t) \times [0, W_t)$, our neural matching field $M \in \mathbb{R}^{H_s \times W_s \times H_t \times W_t}$ is computed as

$$M(\mathbf{p}) = f_\theta(\gamma(\mathbf{p}), \phi(C, \mathbf{p})) \in [0, 1], \tag{4}$$

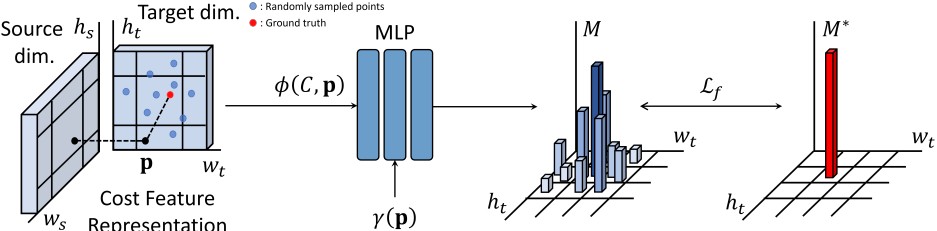

Figure 4: **Overview of neural matching field optimization:** Given an encoded cost, we randomly sample coordinates from uniform distribution. The random coordinates and ground-truth coordinate are then processed altogether to obtain the matching scores and the cross-entropy loss is computed for the training signal.

where $\gamma(\mathbf{p})$ is an encoded point of $\mathbf{p}$, and $\phi(C, \mathbf{p})$ denotes the cost feature vector at $\mathbf{p}$ extracted from coarse cost volume $C$. A possible way for the architecture design of function $f_\theta$ is first concatenating the two inputs $\gamma(\mathbf{p})$ and $\phi(C, \mathbf{p})$, and then passing them through the fully-connected network. This approach, however, may impose memory intensive batch normalization operation [44]. To address this, we adopt the architecture of [52], where $\phi(C, \mathbf{p})$ is added to the input features of each fully-connected block. In addition, to guarantee the value of 4D matching cost fields to be lied within the range from $0$ to $1$, we use a sigmoid function [17] at the end of the networks.

## 4.2 Cost Embedding Network

Our assumption is that formulating the function $f_\theta(\cdot)$ without any condition may be challenged when representing complicated and high-dimensional continuous field. We address this by introducing cost embedding network where the raw cost volume is further embedded into a cost feature volume $\phi(C, \mathbf{p})$, which is used as a guidance for establishing high-precision matching field through the following fully-connected network $f_\theta(\cdot)$.

Motivated by recent works [8, 22] that aggregate matching costs for better correspondence hypothesis, we further embed the raw cost volume through the global receptive fields of self-attention layer [78, 12]. Although these representations are explicitly encoded from all pixels of a cost volume, the absence of operations that impart inductive bias, *i.e.,* translation equivariance by convolutions or relative positioning bias, may yield representations with errors. To this end, we combine Transformer architecture [78] with convolution operator to compensate the lack of inductive bias, allowing local and global integration of matching cues by encapsulating the local contexts and imparting them to all pixels via self-attention. Concretely, before providing matching cost to the function $f_\theta$, the 4D raw cost volume $C$ is embedded into 5D cost feature volume $C' \in \mathbb{R}^{H_s \times W_s \times H_t \times W_t \times K}$ with $K$ channels which is still defined at low resolution. We then use a quadlinear interpolation on $C'$ for a query point $\mathbf{p}$ to a cost feature vector $\phi(C, \mathbf{p}) \in \mathbb{R}^{K \times 1}$.

## 4.3 Training

As shown in Fig. 4, to train the networks, we use a ground-truth keypoint pair $\{\mathbf{x}, \mathbf{x}'\}$ between an input image pair in a manner that if a query point $\mathbf{p} = [\mathbf{x}, \mathbf{y}]$ is classified as the ground-truth correspondence, *i.e.,* $\mathbf{y} = \mathbf{x}'$, the network output should be encouraged to be 1, and 0 otherwise. We formulate this as a classification problem, and thus we apply cross-entropy loss to learn to predict the correctness of a correspondence for a query $\mathbf{x}$ in the source image among sampled negative keypoints $\mathbf{y}$ in the target image (where $\mathbf{y} \neq \mathbf{x}'$) and that of ground-truth $\mathbf{x}'$. Even though the better negative sampling techniques, e.g., hard negative mining [27, 24], can be used, in experiments, we simply adopt random sampling from uniform distribution as negative samples, as background clutters or extreme geometric variations inherently present across semantic correspondence datasets [15, 16, 49] would contribute to robust representation learning. Formally, the loss function is defined as follows:

$$\mathcal{L}_f = -\sum_{k=1}^{K} M_k^* \log(M(\mathbf{p}_k)), \tag{5}$$

where $\mathbf{p}_k$ denotes $k$-th query samples for $k \in \{1, ..., K\}$, and $M_k^*$ is a ground-truth matching score, e.g., $M_k^* = 1$ if $\mathbf{p}_k$ is a ground-truth keypoint pair, and $0$ otherwise.

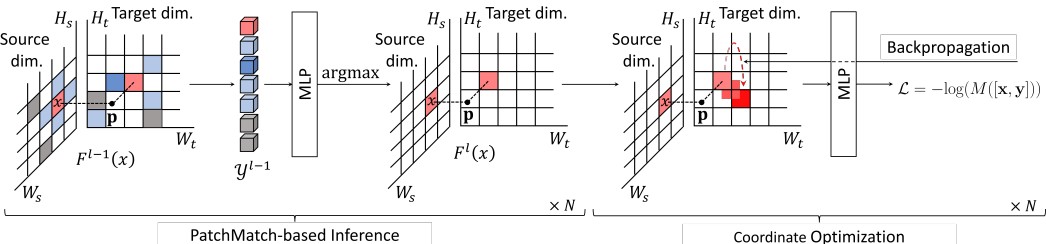

Figure 5: **Illustration of the proposed PatchMatch and coordinate optimization:** With the learned neural matching field, the proposed PatchMatch injects explicit smoothness and reduces the search range. The subsequent optimization strategy searches for a location that maximizes the score of MLP.

Although this would provide sufficient supervisory signal, we found it is beneficial to provide an additional explicit supervisory signal for learning better cost features, which positively affect our PatchMatch [1]-based inference strategy as will be further detailed in Sec. 4.4. To this end, we use end-point-error [8] between the predicted keypoints using the cost feature representations $\phi(C, \mathbf{p})$ directly and the ground-truth keypoints. Concretely, we obtain a channel-wise average pooled cost feature volume $V = \mathrm{avgpool}(\phi(C, \mathbf{p})) \in \mathbb{R}^{H_s \times W_s \times H_t \times W_t}$ and compute $F_{\mathrm{pred}}$ by applying soft argmax function to $V$. We then calculate the Euclidean distance between the ground-truth flow map computed using the ground-truth keypoints and the predicted flow map as

$$\mathcal{L}_c = \|F_{\mathrm{gt}} - F_{\mathrm{pred}}\|_2. \tag{6}$$

Combining with Eq. 6, we define the final objective function $\mathcal{L}_{\mathrm{total}}$ with balancing weights $\lambda_f$ and $\lambda_c$: $\mathcal{L}_{\mathrm{total}} = \lambda_f \mathcal{L}_f + \lambda_c \mathcal{L}_c$.

## 4.4 Inference

**PatchMatch-based Sampling.** At the inference stage, we aim to find a dense correspondence field $F(\mathbf{x})$ by leveraging the trained network to determine the correct correspondence for each query $\mathbf{p} = [\mathbf{x}, \mathbf{y}]$. However, searching the best match for each coordinate $\mathbf{x}$ over all possible matching candidates $\mathbf{y}$ in exhaustive manner results in $H_s \times W_s \times H_t \times W_t$ number of feed-forward per sample, which is an extremely time consuming and computationally intensive.

To alleviate the issue, we propose PatchMatch [1]-based sampling. PatchMatch [1] runs a sequence of propagation and update steps to reduce a search space. We use the learned NeMF $f_\theta$ as a scoring function to determine the correspondence. To overcome its time-consuming process induced by serial processing inherited from PatchMatch [1], we propose a GPU-friendly PatchMatch optimization that performs propagation and update in a parallel manner.

Specifically, for initialization, we utilize an average pooled cost feature volume $V$ introduced in Section 4.3 . Note that the objective of Eq. 6 directly connects to the initialization step of this approach. This implies that the better cost feature representations would help to obtain a better initialization for PatchMatch-based inference. Then for a query $\mathbf{x}$, we sample a set of candidate correspondences by considering adjacent pixels such that $\mathcal{Z}^{l-1} = \{F^{l-1}(\mathbf{z})\}_\mathbf{z}$ for adjacent pixels $\mathbf{z}$ at $(l-1)$-th iteration. In addition, a few random points sampled from a uniform distribution are used to augment $\mathcal{Z}^{l-1}$ such that $\mathcal{Y}^{l-1} = \bigcup \left( \mathcal{Z}^{l-1}, \{\mathbf{y}\} \right)$ for randomly sampled pixels $\mathbf{y}$ where $\bigcup$ denotes an union of the sets. Then the correspondence fields are undated by considering the set of matching candidate $\mathcal{Y}^{l-1}$ such that

$$F^l(\mathbf{x}) = \mathrm{argmax}_{\mathbf{y} \in \mathcal{Y}^{l-1}}(M([\mathbf{x}, \mathbf{y}])). \tag{7}$$

This process is iterated until the convergence. In practice, this candidate selection and scoring run in parallel for every target pixel. This makes the inference process efficient and GPU-friendly in the original resolution, compared to serial propagation and update in [30].

**Coordinate Optimization.** Although the proposed PatchMatch-based inference strategy could prevent exhaustive searching by effectively sampling the search range for determining the correct correspondence for each pixel, this may degrade the performance due to several reasons including insufficient number of iterations that may result in a sub-optimal solution and a limited search range that provides relatively fewer candidates for consideration. To address this issue, we provide a means

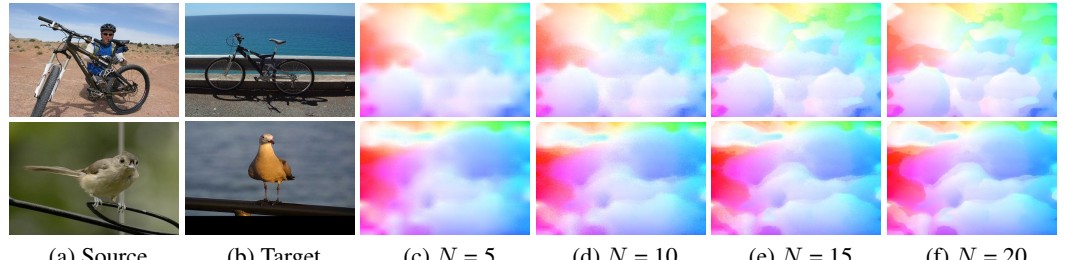

|   (a) Source   |   (b) Target   |   (c) $N = 5$   |   (d) $N = 10$   |   (e) $N = 15$   |   (f) $N = 20$   |

Figure 6: **Visualization of flow maps for different** $N$ **iterations:** (a) source image, (b) target image. As the number of iteration we set increases along (c), (d), (e) and (f) at inference phase, NeMF with trained MLP predicts more precise matching fields by PatchMatch-based sampling and coordinate optimization. Note that different colors of flow maps indicate directions and magnitudes of the flows.

to reduce the potential erroneous inference by adopting test-time optimization strategy that directly optimize coordinates **y** that maximizes the correctness of the correspondence using the learned the network $f_\theta$.

Concretely, because the network $f_\theta$ is naturally differentiable, as shown in Fig. 5, we use a gradient descent to optimize the target coordinate **y** in the direction of decreasing the negative log likelihood of the matching score, with respect to the corresponding source coordinate **x**. Formally, the coordinate optimization is performed by iterative updates which can be formulated as:

$$
\begin{aligned}
\mathcal{L} &= -\log(M([\mathbf{x}, \mathbf{y}])), \\
\mathbf{y} &:= \mathbf{y} - \alpha \nabla_{\mathbf{y}} \mathcal{L},
\end{aligned}
\tag{8}
$$

where $\alpha$ denotes a step size. Any advanced optimizer can also be used for improved optimization [60, 39]. Note that the source coordinate is not updated during this optimization. With the proposed coordinate optimization, we combine with PatchMatch-based sampling to establish a final correspondence field as shown in Fig. 5. Each iteration number is defined as $N$. As exemplified in Fig. 6, NeMF predicts more precise matching fields by PatchMatch-based sampling and coordinate optimization as evolving iterations.

Note that the key difference of this test-time optimization to that of DMP [19] is that we optimize the coordinates to correct themselves to find a better correspondence with by leveraging the already learned network, while DMP optimizes the parameters of the networks.

## 5 Experiments

### 5.1 Implementation Details

For backbone feature extractor, we use ResNet-101 [18] pre-trained on ImageNet [11]. We use the feature maps resized to $16 \times 16$ for constructing a coarse cost volume. For the cost embedding network, we build upon [8] and its implementations. We implemented our network using PyTorch [56], and AdamW [41] optimizer with an initial learning rate of 3e−5. We set $N = 10$ for both PatchMatch and coordinate optimizations, and learning rate of 3e−4 is used for coordinate optimization. Additional details are provided in the supplemenatry material.

### 5.2 Experimental Settings

**Datasets.** We use three benchmarks, which include SPair-71k [49], PF-PASCAL [16] and PF-WILLOW [15], to evaluate the effectiveness of the proposed method. SPair-71k [49] provides total 70,958 image pairs, PF-PASCAL [16] contains 1,351 image pairs from 20 categories, and PF-WILLOW [15] contains 900 image pairs from 4 categories. Each dataset contains ground-truth annotations, which we use them for evaluation and training.

**Evaluation Metric.** For the evaluation metric, we employ a percentage of correct keypoints (PCK), which is computed as the ratio of estimated keypoints within the threshold from ground-truths to the total number of keypoints. Assume a predicted keypoint $k_{\text{pred}}$ and a ground-truth keypoint $k_{\text{GT}}$, the number of correctly predicted keypoints are counted, and the condition for deciding the

Table 1: **Quantitative evaluation on standard benchmarks [49, 15, 16, 75]:** Higher PCK is better. The best results are in bold, and the second best results are underlined. All results are taken from the papers. *Eval. Reso.: Evaluation Resolution, Flow Reso.: Flow Resolution.*

| Methods | Eval. Reso. | Flow Reso. | SPair-71k [49] PCK @ $\alpha_{bbox}$ | | | | PF-PASCAL [16] PCK @ $\alpha_{img}$ | | | | PF-WILLOW [15] PCK @ $\alpha_{bbox-kp}$ | | | |
|---|---|---|---|---|---|---|---|---|---|---|---|---|---|---|
| | | | 0.01 | 0.03 | 0.05 | 0.1 | 0.01 | 0.03 | 0.05 | 0.1 | 0.01 | 0.03 | 0.05 | 0.1 |
| CNNGeo [62] | ori | - | - | - | - | 20.6 | - | - | 41.0 | 69.5 | - | - | 36.9 | 69.2 |
| A2Net [71] | - | - | - | - | - | 22.3 | - | - | 42.8 | 70.8 | - | - | 36.3 | 68.8 |
| WeakAlign [63] | ori | - | - | - | - | 20.9 | - | - | 49.0 | 74.8 | - | - | 37.0 | 70.2 |
| RTNs [29] | - | - | - | - | - | 25.7 | - | - | 55.2 | 75.9 | - | - | 41.3 | 71.9 |
| SFNet [32] | 288/ori | 20 | - | - | - | - | - | - | 53.6 | 81.9 | - | - | 46.3 | 74.0 |
| PARN [25] | - | - | - | - | - | - | - | - | 26.8 | 49.1 | - | - | - | - |
| PMD [35] | - | 20 | - | - | - | 37.4 | - | - | - | 90.7 | - | - | - | 75.6 |
| PMNC [31] | 400 | - | - | - | - | 50.4 | - | - | **82.4** | 90.6 | - | - | - | - |
| MMNet [82] | 224×320 | - | - | - | - | 40.9 | - | - | 77.6 | 89.1 | - | - | - | - |
| DCC-Net [21] | 240/ori/- | - | - | - | - | - | - | - | 55.6 | 82.3 | - | - | 43.6 | 73.8 |
| HPF [48] | max 300 | - | - | - | - | 28.2 | - | - | 60.1 | 84.8 | - | - | 45.9 | 74.4 |
| GSF [26] | - | - | - | - | - | 36.1 | - | - | 65.6 | 87.8 | - | - | 49.1 | **78.7** |
| ANC-Net [34] | 240 | 15 | - | - | - | - | - | - | - | 86.1 | - | - | - | - |
| NC-Net [65] | 240/ori/- | 15 | - | - | - | 20.1 | - | - | 54.3 | 78.9 | - | - | 33.8 | 67.0 |
| DHPF [50] | 240 | 15 | - | - | - | 37.3 | - | - | 75.7 | 90.7 | - | - | - | 71.0 |
| CHM [47] | 240 | 15 | - | - | - | 46.3 | - | - | 80.1 | 91.6 | - | - | - | 69.6 |
| CATs [8] | 256 | 16 | 2.3 | 13.8 | 27.7 | 49.9 | 7.7 | 49.9 | 75.4 | 92.6 | 2.9 | 20.4 | 40.7 | 69.0 |
| **NeMF** | ori | ori | **3.2** | **19.5** | **34.2** | **53.6** | **18.6** | **61.6** | 80.6 | **93.6** | **3.8** | **25.4** | **60.8** | 75.0 |

Table 2: **Per-class quantitative evaluation on SPair-71k [49] benchmark.**

| Methods | aero. | bike | bird | boat | bott. | bus | car | cat | chai. | cow | dog | hors. | mbik. | pers. | plan. | shee. | trai. | tv | all |
|---|---|---|---|---|---|---|---|---|---|---|---|---|---|---|---|---|---|---|---|
| CNNGeo [62] | 23.4 | 16.7 | 40.2 | 14.3 | 36.4 | 27.7 | 26.0 | 32.7 | 12.7 | 27.4 | 22.8 | 13.7 | 20.9 | 21.0 | 17.5 | 10.2 | 30.8 | 34.1 | 20.6 |
| WeakAlign [63] | 22.2 | 17.6 | 41.9 | 15.1 | 38.1 | 27.4 | 27.2 | 31.8 | 12.8 | 26.8 | 22.6 | 14.2 | 20.0 | 22.2 | 17.9 | 10.4 | 32.2 | 35.1 | 20.9 |
| NC-Net [65] | 17.9 | 12.2 | 32.1 | 11.7 | 29.0 | 19.9 | 16.1 | 39.2 | 9.9 | 23.9 | 18.8 | 15.7 | 17.4 | 15.9 | 14.8 | 9.6 | 24.2 | 31.1 | 20.1 |
| HPF [48] | 25.2 | 18.9 | 52.1 | 15.7 | 38.0 | 22.8 | 19.1 | 52.9 | 17.9 | 33.0 | 32.8 | 20.6 | 24.4 | 27.9 | 21.1 | 15.9 | 31.5 | 35.6 | 28.2 |
| SCOT [40] | 34.9 | 20.7 | 63.8 | 21.1 | 43.5 | 27.3 | 21.3 | 63.1 | 20.0 | 42.9 | 42.5 | 31.1 | 29.8 | 35.0 | 27.7 | 24.4 | 48.4 | 40.8 | 35.6 |
| DHPF [50] | 38.4 | 23.8 | 68.3 | 18.9 | 42.6 | 27.9 | 20.1 | 61.6 | 22.0 | 46.9 | 46.1 | 33.5 | 27.6 | 40.1 | 27.6 | 28.1 | 49.5 | 46.5 | 37.3 |
| CHM [47] | 49.1 | 33.6 | 64.5 | 32.7 | 44.6 | 47.5 | 43.5 | 57.8 | 21.0 | 61.3 | 54.6 | 43.8 | 35.1 | 43.7 | 38.1 | 33.5 | 70.6 | 55.9 | 46.3 |
| MMNet [82] | 43.5 | 27.0 | 62.4 | 27.3 | 40.1 | 50.1 | 37.5 | 60.0 | 21.0 | 56.3 | 50.3 | 41.3 | 30.9 | 19.2 | 30.1 | 33.2 | 64.2 | 43.6 | 40.9 |
| PMNC [30] | 54.1 | 35.9 | 74.9 | 36.5 | 42.1 | 48.8 | 40.0 | **72.6** | 21.1 | 67.6 | 58.1 | 50.5 | 40.1 | **54.1** | 43.3 | 35.7 | 74.5 | 59.9 | 50.4 |
| CATs [8] | 52.0 | 34.7 | 72.2 | 34.3 | 49.9 | 57.5 | 43.6 | 66.5 | 24.4 | 63.2 | 56.5 | 52.0 | 42.6 | 41.7 | 43.0 | 33.6 | 72.6 | 58.0 | 49.9 |
| **NeMF** | **55.6** | **37.2** | **76.2** | **36.9** | **54.1** | **62.1** | **47.5** | 70.5 | **26.2** | **67.6** | **59.3** | **57.1** | **48.0** | 40.2 | 42.1 | **36.7** | **80.7** | **66.1** | **53.6** |

correctness is defined as follows: $d(k_{\text{pred}}, k_{\text{GT}}) \leq \alpha \cdot \max(H, W)$, where $d(\cdot)$ and $\alpha$ denote Euclidean distance and a threshold. When we evaluate on PF-PASCAL, we use $\alpha_{\text{img}}$ following other works [16, 48, 50, 8], SPair-71k and PF-WILLOW with $\alpha_{\text{bbox}}$; $H$ and $W$ denote height and width of the object bounding box or entire image, respectively.

## 5.3 Matching Results

To ensure a fair comparison, the model evaluated on SPair-71k [49] is trained on training split of SPair-71k [49] and the model evaluated on PF-PASCAL [16] and PF-WILLOW [15] is trained on training split of PF-PASCAL [16].

The results are summarized in Table 1 and the qualitative results are shown in Fig. 7. We note the resolution which the method is evaluated, since [9, 77] observe that the resolution of images affect the PCK performance, and the resolution of which the method outputs the correspondence field. It is shown that NeMF achieves competitive performance or even attains state-of-the-art performance for several alpha thresholds. More concretely, for lower alpha thresholds, we tend to achieve higher PCK compared to other works. This implies that the existing works, which rely on interpolation techniques that prevent from fine-grained matching due to the use of matching field defined at low resolution, may have suffered from the large resolution gap between the predicted flow and that of ground-truth. For example, CATs [8] processes the cost volume at $16^4$ and infers a flow map at this resolution. On the contrary, NeMF avoids this by implicitly representing a matching field at higher resolution, demonstrating its advantageous approach.

## 5.4 Ablation Study

In this section, we conduct ablation study to investigate the effectiveness of different configurations for cost embedding network and effectiveness of the proposed inference strategies. We train the networks on the training split of SPair-71k [49] and evaluated on the test split.

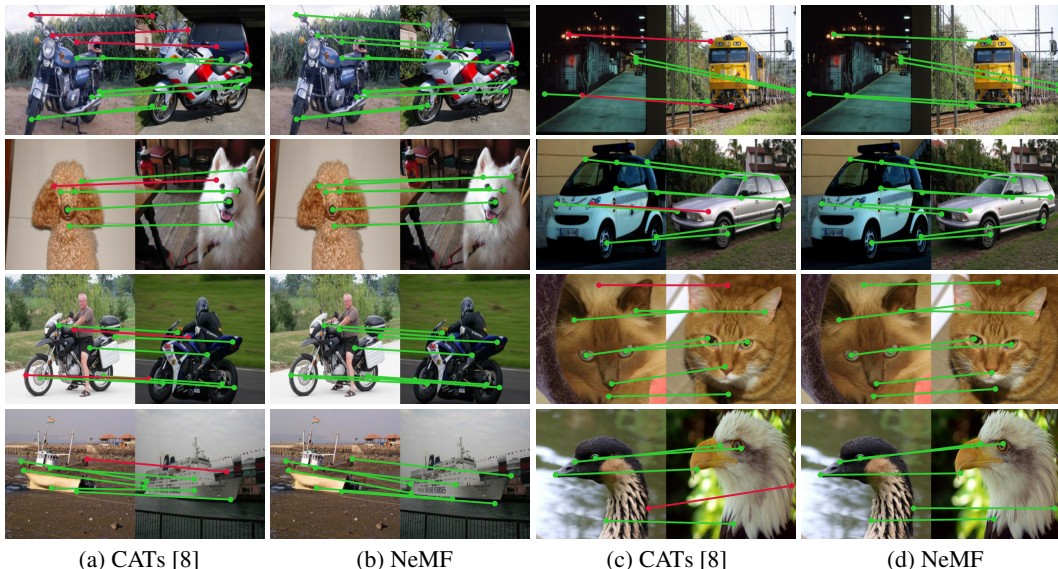

| (a) CATs [8] | (b) NeMF | (c) CATs [8] | (d) NeMF |

Figure 7: **Qualitative results on PF-PASCAL [16]:** keypoint transfer results by (a), (c) CATs [8] and (b), (d) NeMF. Green and red line denote correct and wrong prediction ($\alpha_{\mathrm{img}} = 0.1$), respectively. Note that correspondences are estimated at the original resolutions of iamges.

**Different Cost Feature Representation.** Table 3 summarizes the comparison between the effectiveness of cost feature representations learned from different configurations of cost embedding network. In this ablation study, we compare four configurations. (**I**) shows the results for exploiting raw cost volume defined at coarse level. From (**II**) to (**IV**), we show the effectiveness of extracting cost features via convolutions, self-attention layers and integration of both, respectively.

We observe that simply leveraging a raw cost volume struggles to learn a complicated matching field, as it does not provide a sufficient structural or detailed information among pixel-wise similarities. As the cost embedding network is introduced to learn feature representations, the performance is dramatically boosted, and our approach clearly helps to attain the best performance by learning more powerful representations than (**II**) and (**III**).

Table 3: **Cost feature representation.**

|  | Components | SPair-71k [49] PCK @ $\alpha_{\mathrm{bbox}}$ | | | | |
|---|---|---|---|---|---|---|
|  |  | 0.01 | 0.03 | 0.05 | 0.1 | 0.15 |
| (**I**) | Coarse cost volume | 0.3 | 2.3 | 5.8 | 15.5 | 25.7 |
| (**II**) | Conv. | 2.2 | 14.6 | 28.3 | 48.9 | 59.4 |
| (**III**) | Self-attention | 2.5 | 16.4 | 30.6 | 51.7 | 61.7 |
| (**IV**) | Conv. + self-attention | 3.2 | 19.5 | 34.2 | 53.6 | 63.3 |

**Inference Strategies.** In this ablation study, we aim to show a quantitative comparison between different strategies at the inference phase. Table 4 summarizes the results. Note that we included (**I**) to highlight that at the inference

Table 4: **Inference strategy.** *TL* denotes Too Long.

|  | Components | SPair-71k [49] PCK @ $\alpha_{\mathrm{bbox}}$ | | | | | Average run-time per sample [s] |
|---|---|---|---|---|---|---|---|
|  |  | 0.01 | 0.03 | 0.05 | 0.1 | 0.15 |  |
| (**I**) | Exhaustive infer. | *TL* | *TL* | *TL* | *TL* | *TL* | > 300k |
| (**II**) | PatchMatch-based infer. | 1.1 | 7.7 | 15.3 | 31.4 | 41.9 | 7.75 |
| (**III**) | (**II**) + coordinate opt. | 3.2 | 19.5 | 34.2 | 53.6 | 63.3 | 8.20 |

phase, the evaluation on a pair of images with original resolution, for example, would take approximately more than 300k seconds. This clearly shows the infeasibility of adopting naïve inference strategy.

From (**II**) to (**III**), we observe an apparent performance boost, which demonstrates that the proposed test-time coordinate optimization helps to correct the coordinates for finding better correspondences. However, this approach has a downside. Applying coordinate optimization inevitably increases the time taken for the inference, which is a typical limitation of test-time optimization. However, 0.5 second is a minor sacrifice for a better performance. Note that further improvement could be made by adopting better optimizing strategy, *i.e.,* learning rate, search range or optimizer.

**Computational Complexity.** Although the proposed inference strategy enables significantly reduced time for establishing correspondence field between a pair of images, in practice, we observe that assuming we set $N = 10$, the time taken at the inference phase for a single sample is approximately 8-9 seconds on a single GPU Geforce RTX 3090, which prevents from a real-time inference. This is an apparent limitation of the proposed approach, but we refer the readers to supplementary material where we show that without affecting the performance, the memory consumption and run-time can be controlled.

In addition, we also provide experimental results that demonstrates the efficiency of the proposed approach in comparison to existing works, which is summarized in Table 5. Let us assume that we are representing cost volumes of four different resolutions, *e.g.,* 16, 32, 64 and 128. At the training phase, unlike other works that inevitably consume more memory as the resolution increases, the proposed approach successfully deviates from it thanks to the proposed training strategy. Furthermore, at the inference phase, we observe that the proposed approach has an advantage over other methods. Although NeMF may suffer from relatively larger computation and memory consumption than CATs [8] and CHM [47] when the resolution is low, it has an advantage when the resolution is high, allowing the network to exploit highly accurate cost volume with relatively less memory consumption.

Table 5: **Memory Comparison.** *OOM : Out of Memory*

| Method | Train | Inference | $16^4$ | $32^4$ | $64^4$ | $128^4$ |
|---|---|---|---|---|---|---|
| CHM [47] | ✓ | ✗ | 708 | 1,538 | OOM | OOM |
| | ✗ | ✓ | 371 | 433 | OOM | OOM |
| CATs [8] | ✓ | ✗ | 454 | 3,523 | OOM | OOM |
| | ✗ | ✓ | 188 | 302 | 1882 | OOM |
| **NeMF** | ✓ | ✗ | 4,205 | 4,205 | 4,205 | 4,205 |
| | ✗ | ✓ | 1,528 | 1,528 | 2,443 | 6,309 |

## 6  Conclusion

In this paper, we proposed a novel INR-based architecture, called neural matching fields (NeMF), that implicitly represent a 4D matching field to find high-precision correspondences. This method proposed an architecture and training and inference procedures targeted to handle complicacy and high-dimensionality of a matching field that acts as major hindrances. Specifically, we embed the raw cost volume with convolutions and Transformer to obtain local and global integration of matching cues to handle the complicacy, and sampling-based training and inference procedure to handle the high-dimensionality. We have shown that the proposed method attains state-of-the-art performance on several benchmarks for semantic correspondence. We also conducted an extensive ablation study to validate our choices.

## Broader Impact

Our implicit representation of cost volume may be beneficial for other domains that utilize a correlation map, which include semantic segmentation [67, 75, 46], object detection [36], and image editing [37]. It can help to boost the performance by preserving the fine-detailed information within the cost volume. However, as the proposed approach aims to implicitly represent the cost volume, on its own, it is not feasible to use for a malicious purpose.

**Acknowledgements.** This research was supported by the MSIT, Korea (IITP-2022-2020-0-01819, ICT Creative Consilience program, No. 2020-0-00368, A Neural-Symbolic Model for Knowledge Acquisition and Inference Techniques), and National Research Foundation of Korea (NRF-2021R1C1C1006897).

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
