# Neural Matching Fields: Implicit Representation of Matching Fields for Visual Correspondence
# - Supplementary Materials -

In this document, we provide more implementation details, analysis and psuedo-code of NeMF and more results on SPair-71k [6], PF-PASCAL [3], and PF-WILLOW [2].

## Appendix A. More Implementation Details

**Cost Embedding Network Details.** The cost embedding network is based on CATs [1]. More specifically, instead of utilizing hyperpixels, we use the feature maps of last index at each pyramidal layers of ResNet-101. Then we resize their spatial resolutions to $16 \times 16$ using 4D convolutions and compute correlation maps. Then we feed them into subsequent Transformer [1] by treating the level dimension as channel, which in this case is 4, and obtain a cost feature volume that has a shape of $\mathbb{R}^{16 \times 16 \times 16 \times 16 \times 16}$.

**Training Details.** We use AdamW [4] with learning rate $3\mathrm{e}^{-5}$. For the MLP architecture, we compose with 3 blocks, which the each block consists of 2 fully connected networks followed by ReLU activation and a residual connection. For uniform sampling, we sample both directions of cost volume and use them to compute the final loss. We use negative log likelihood function with temperature $\tau = 0.07$ for computing the loss between the predicted correspondence and the ground-truth correspondence. We use EPE loss additionally using the predicted flow and the ground-truth flow from cost embedding network. Balancing factors $\lambda_f$ and $\lambda_c$ are set to 1. For the frequency of the positional encoding $L$ for coordinates, we set as $L = 10$. We use PyTorch3D [8] to encode the coordinates.

## Appendix B. Controlling computational complexity

Additionally, we emphasize that with only a negligible amount of influence on the performance, we can reduce computational burden and memory consumption at inference phase by only optimizing the coordinates of interests used for evaluation at coordinate optimization phase and tuning the batch-size of the input coordinates to the PatchMatch-based sampling, which can determine the memory consumption and run-time. Assuming a set of keypoints are for querying is available, we can optimize only the coordinates of the keypoint which we want to find its corresponding keypoint at target image. This way, we can significantly reduce the run-time. The results are shown in Table 1. For this experiment, we assumed NeMF is representing cost volume of size 128 to show the memory comparison that will be presented below this paragraph. From the results, we observe negligible memory change, but significant reduction in run time when only the keypoints of interest are optimized.

Also, by reducing the batch-size of the input coordinates to the PatchMatch-based sampling we can also control the memory consumption. To this end, we conduct a simple experiment and report the run-time and memory consumption with varying batch size. The results are shown in Table 2. This table shows that tuning the batch-size can reduce the memory consumption by sacrificing run-time, meaning that users can choose to infer with high/low memory and fast/slow run-time.

| Inference strategy | Run-time [s/img] | Memory [MiB] |
|---|---|---|
| PatchMatch Only | 1.42 | 6307 |
| PatchMatch + Optimize all coordinates | 2.21 | 6309 |
| PatchMatch + Optimize only keypoints | 1.65 | 6308 |

Table 1: **Computation complexity comparison.**

| Batch size | Run-time [s/img] | Memory [MiB] |
|---|---|---|
| 100000 | 1.65 | 6308 |
| 50000 | 2.27 | 4301 |
| 25000 | 4.43 | 2730 |
| 10000 | 9.18 | 1789 |

Table 2: **Computation complexity comparison.**

## Appendix C. Additional Results

**More Qualitative Results.** We provide more comparison of CATs and other state-of-the-art methods on SPair-71k [6] in Fig. 1, PF-PASCAL [3] in Fig. 2, and PF-WILLOW [3] in Fig. 3. We also present visualization of matching fields on SPair-71k [6] in Fig. 4.

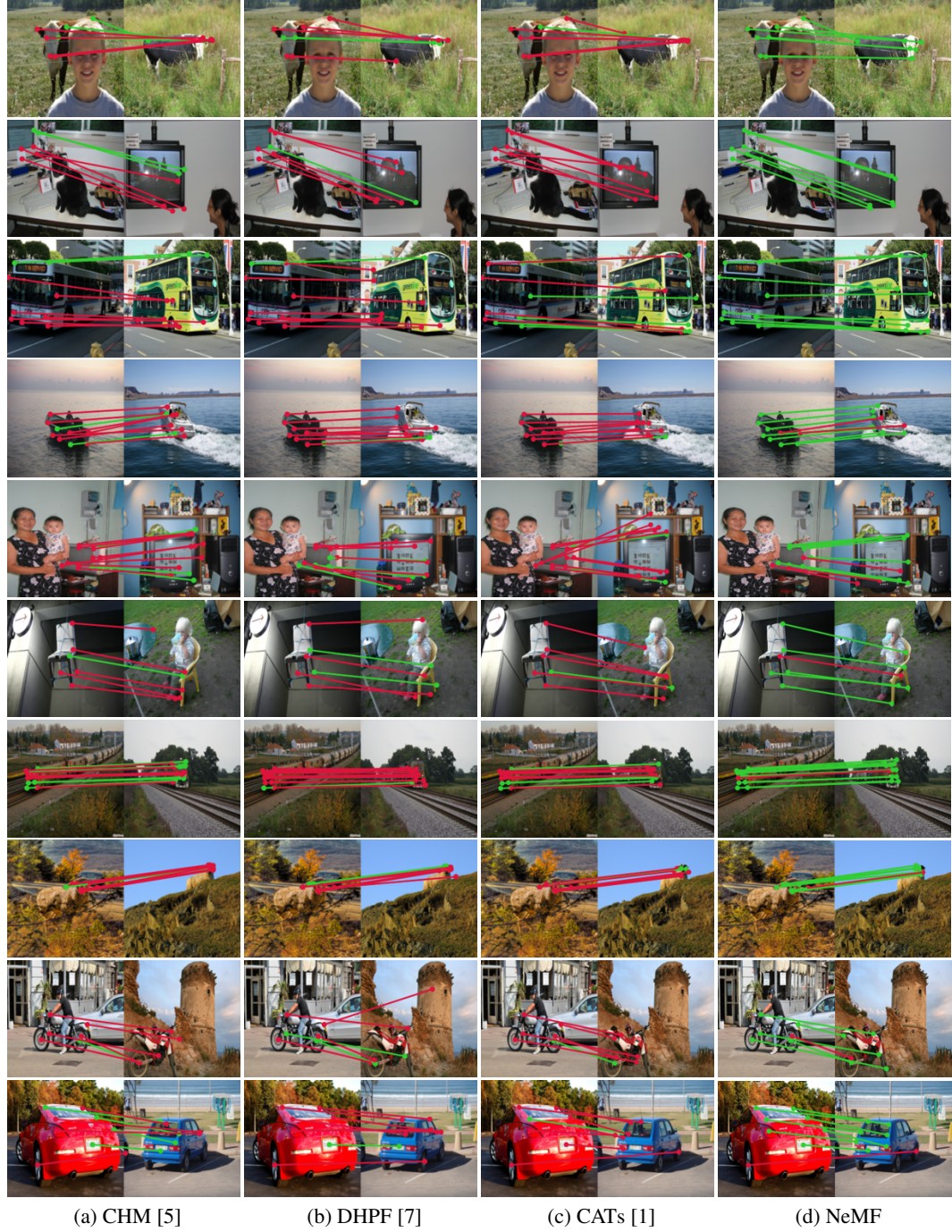

|(a) CHM [5]|(b) DHPF [7]|(c) CATs [1]|(d) NeMF|

Figure 1: **Qualitative results on SPair-71k [6]:** keypoints transfer results by (a) CHM [5], (b) HPF [7], and (c) CATs [1], and (d) NeMF. Note that green and red line denotes correct and wrong prediction, respectively, with respect to the ground-truth.

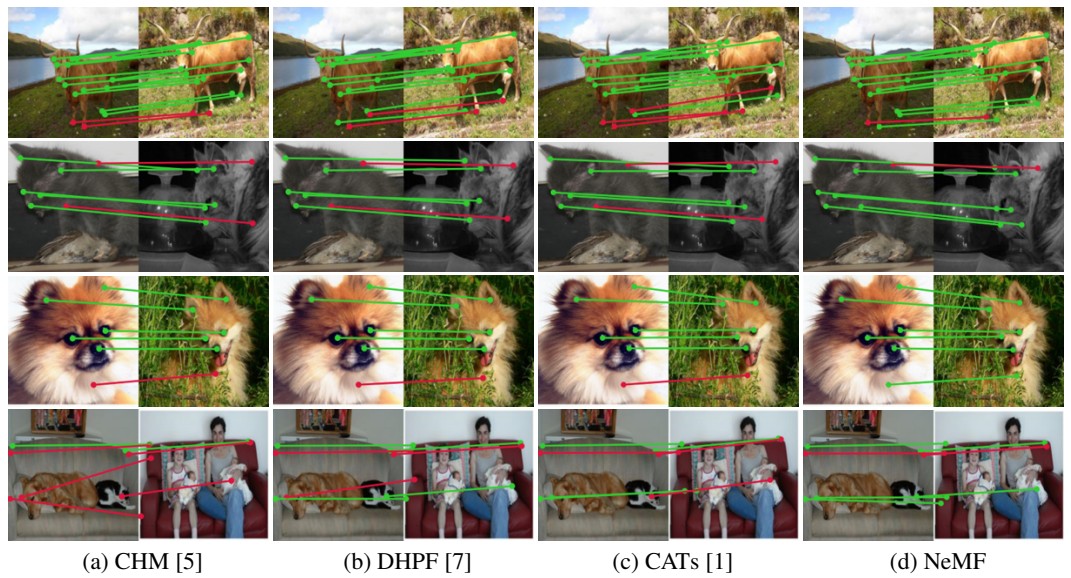

|         |         |         |         |
|:-------:|:-------:|:-------:|:-------:|
| (a) CHM [5] | (b) DHPF [7] | (c) CATs [1] | (d) NeMF |

Figure 2: **Qualitative results on PF-PASCAL [3]**

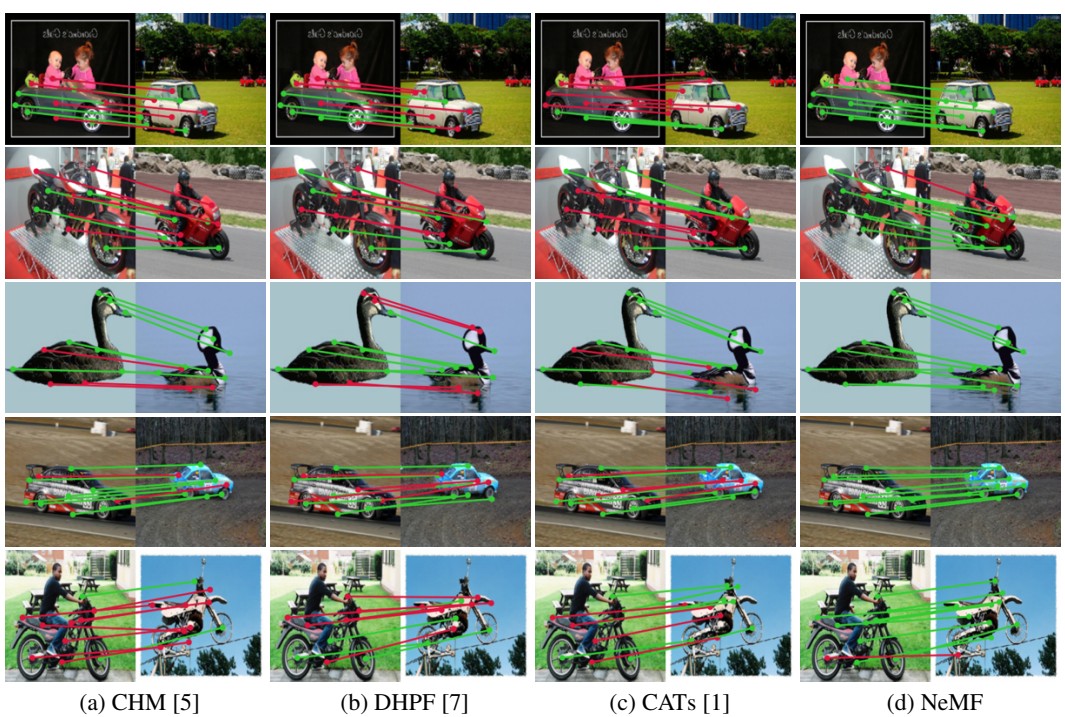

|         |         |         |         |
|:-------:|:-------:|:-------:|:-------:|
| (a) CHM [5] | (b) DHPF [7] | (c) CATs [1] | (d) NeMF |

Figure 3: **Qualitative results on PF-WILLOW [2]**

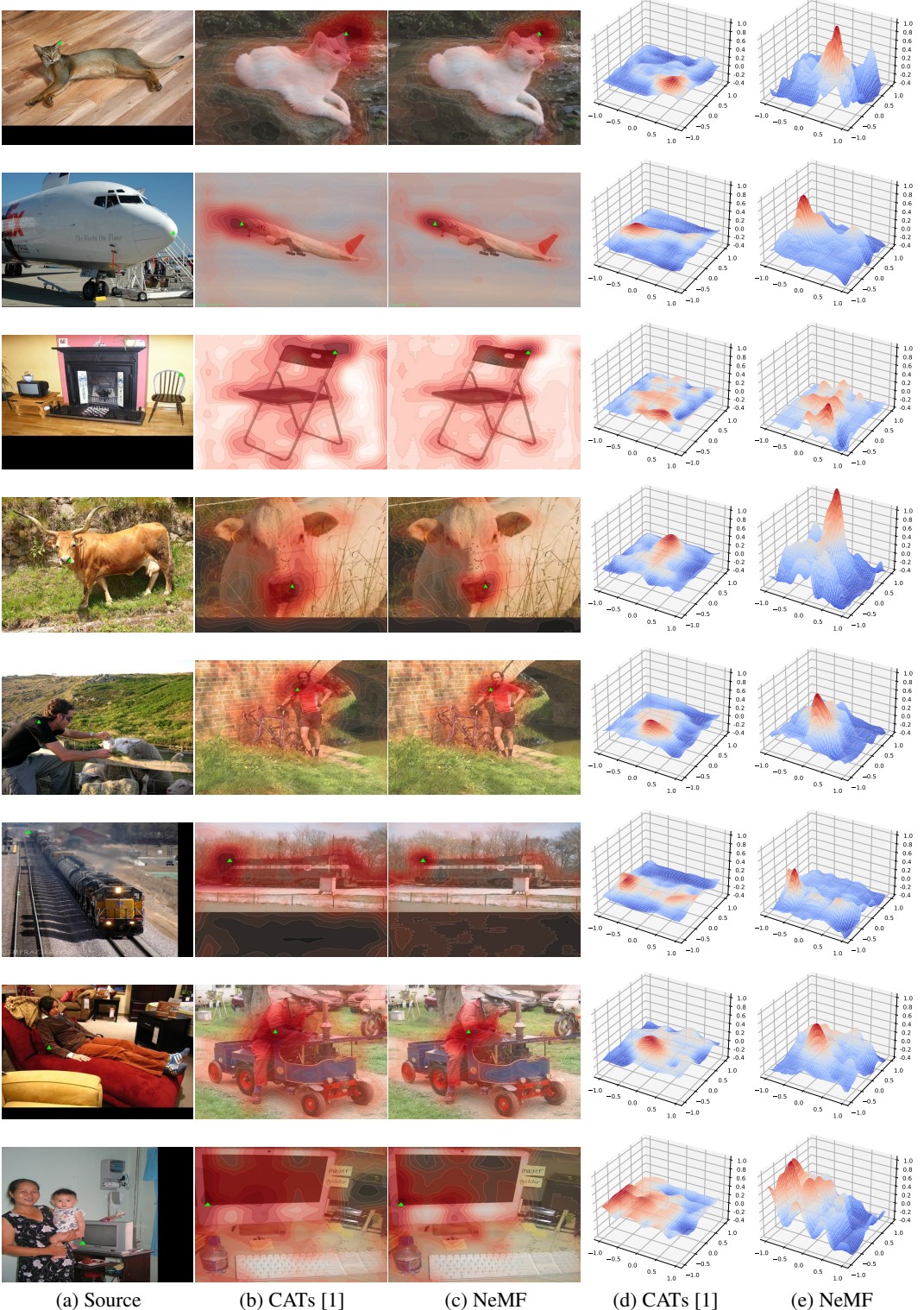

(a) Source      (b) CATs [1]      (c) NeMF      (d) CATs [1]      (e) NeMF

Figure 4: **Visualization of matching fields on SPair-71k [6]:** (a) source image, where the keypoint is marked as green triangle, (b), (c) 2D contour plots of cost by CATs [1] and the NeMF (ours), respectively, and (d), (e) 3D visualization of cost by CATs [1] and NeMF, with respect to the keypoint in (a). Note that all the visualizations are smoothed by a Gaussian kernel.