# OpenReview forum: "Neural Matching Fields: Implicit Representation of Matching Fields for Visual Correspondence"
_NeurIPS.cc/2022/Conference — NeurIPS 2022 Accept_

### Official Review · Reviewer_vfUA · 2022-06-30

**Rating:** 6
**Confidence:** 4
**Soundness:** 3 good
**Presentation:** 2 fair
**Contribution:** 3 good

**Summary:**

This paper proposed a novel method based on Implicit neural representation(INR) for semantic correspondence, named neural matching field (NeMF). Also a cost embedding network containing convolution and self-attention layers is established in order to represent complicated and high-dimensional continuous field better. During the inference phase, this paper introduced the patchmatch-based sampling and coordinate optimization which make the NeMF performs better and more efficient. Experiments on several standard benchmarks and ablation study validate the proposed architecture with the full resolution input image pairs.

**Questions:**

Figure 3 illustrates the proposed network architecture but it is not clear, especially the concrete architecture of the cost embedding network.

Figure 6 demonstrates the flow maps for different N iterations, the author should give more explanations on the meanings of different colors.

Figure 7 shows several semantic matches on PF-PASCAL, however, it seems that the results of the third pair in the first two columns and the second pair in the last two columns look almost the same while some correspondences predicted by CATs are marked red. Additionally, the correspondence marked red of the last image pair in the third columns looks different from that reported in the original paper(CATs)

qualitative results of CATs on PF-WILLOW are different from that reported in the original paper (i.e. 50.3 when α=0.05 and 79.2 when α=0.1 )


**Limitations:**

This paper formulates predicting semantic correspondence as a classification problem which lacks of generalization between datasets, Concretely, the network trained on one datasets with fixed number of categories may not performer well on other datasets with unseen categories

**Strengths And Weaknesses:**

Originality: Comparing to existing methods, this paper firstly leverage the implicit neural representation in semantic correspondence.

Quality: Benefiting from INR which is not coupled to spatial resolution, images can be input to the proposed network with their original resolution and SOTA performance is attained with competitive inference time.

Clarity: The paper is somewhat clear, but some important details are missing or unclear

Significance: The paper is likely to have moderate impact on semantic correspondence

---

> ### Author Response · Authors · 2022-08-02
> **Response to reviewer vfUA**
>
> Thanks for the comments. Our responses are in the following. If our responses do not adequately address your concerns, please let us know and we will get back to you as soon as possible.
>
> > **Concrete architecture of the cost embedding network**
>
> We apologize for the absence of such an important detail. Our cost embedding network builds upon CATs [8], while removing appearance affinity modelling and adding convolution prior to Transformer module. We will include this in the supplementary material.
>
> > **Meanings of different colors**
>
> Different colors of flow maps indicate directions and magnitudes of the flows. This visualization has been widely adopted in the optical flow literature, as well as FlowNet2.0 [23]. We will include this in the caption of Figure 6.
>
> >**Figure 7 clarification**
>
> Training and evaluation resolutions have been one of the issues present in the semantic correspondence literature along with the evaluation for PF-WILLOW. The error in PCK threshold for PF-WILLOW is addressed in CHMNet, a journal extension for CHM. The resolution issue is addressed in CATs++ and PWarpCNet. Both works argue that to
> make a fair comparison, the training resolutions can be different
> among works, but the evaluation resolutions should be the same among works. Note that CATs++ explicitly mentions this and PWarpC details its experimental setting in the supplementary material where training resolutions are different and mentions in section 5.3 that the original image size is used during evaluation for a fair comparison.
>
>
> Following this protocol, we reported both quantitative and qualitative results from  images at original resolutions, which would have led to different results to those in the original CATs [8] paper. We will clarify this in Figure 7 caption. Thanks for the comment.
>
> > **PF-WILLOW wrong results**
>
> Recently, CHMNet [45], a journal extension uploaded on arxiv, pointed out that since DHPF [48], the code implementations were wrong for PF-WILLOW, and newly proposed bbox-kp threshold. This is why the quantitative results of CATs on PF-WILLOW (measured using bbox-kp) are different from those of the original CATs paper. After re-computing the results, we obtained:
>
>
> |Model   | PCK 0.01 | PCK 0.03 | PCK 0.05 | PCK 0.1  |
> |---|:---:|:---:|:---:|:---:|
> | CATs  | 2.9  | 20.4  |  40.7 |  69.0 |
> We will replace the values. Our sincere apologies for this.
>
> > **Generalization power**
>
> Conventionally, PF-Willow is evaluated using the model trained on PF-PASCAL, which shows the generalization power of the model. In Table 1, CATs [8], trained and evaluated on PF-PASCAL, achieves 92.6 for PCK at $\alpha= 0.1$, which 1.0 lower performance than NeMF. In contrast, at PF-WILLOW, NeMF significantly outperforms CATs for all $\alpha$ thresholds, indicating its generalization power. Nevertheless, almost all the matching networks including NeMF do not particularly propose or address any means of regularization or loss for zero-shot setting, and this may make them incapable to perform well on unseen categories. Zero-shot semantic matching is indeed an intersting and promising future direction.  Thanks for the comment.

---

### Official Review · Reviewer_iiVf · 2022-07-11

**Rating:** 4
**Confidence:** 5
**Soundness:** 1 poor
**Presentation:** 4 excellent
**Contribution:** 2 fair

**Summary:**

This paper proposes a new matching field generation method called a neural matching field (NeMF). It uses a cost embedding network consisting of convolution and self-attention layers to process the coarse cost volume to obtain cost feature representation. Through such a mechanism, NeMF can realize high-resolution semantic matching. During the training, to reduce the computational cost of the 5-D matching field, NeMF combines the convolution operations with the self-attention operations to get initial matching tensors.  NeMF also incorporates random sampling when calculating the local matching fields.

**Questions:**

1. What is the biggest difference between NeMF and other semantic matching methods based on 5-D matching cost volume?
2. It seems that the output of PatchMatch is used as pseudo labels to guide the training.
3. During the inference, it's unclear why NeMF needs back-propagation.

**Ethics Review Area:**

["I don’t know"]

**Limitations:**

The computational cost of NeMF is relatively high. I don't find any other negative societal impact.

**Strengths And Weaknesses:**

***Strength***
1. This paper is well-organized and can be easily understood by readers. The technical details are introduced clearly.
2. The authors conducted extensive experiments on multiple benchmarks to investigate the effectiveness of different modules and designs in this paper.
3. NeMF aims to get matching fields in high resolutions which can resolve the matching ambiguity during semantic matching.

***Weakness***
1. Some core techniques of this paper have been used in some disparity or optical flow estimation papers. In Figures 3 and 4, the way to construct 5-D matching cost volumes has been widely used in disparity estimation. Besides, although the 5-D matching cost volumes can get richer matching information, it takes in a huge amount of computational cost. When the resolution of the matching field is very large, it's too expensive to process these 5-D matching volumes.
2. In lines 137-138, to get the semantic matching field in high resolution, NeMF uses interpolation methods (as that in MMNet, ICCV 2021) to upscale the matching fields. But such an interpretation will lead to matching ambiguity obviously. Can these ambiguities brought by interpolation methods be eliminated in subsequent modules?
3.  In section 4.3 (lines 180-182), if NeMF uses random sampling, the training will encounter the long-tail problem. Because most of the point pairs will be classified as negative matching, and only very few matched point pairs can be classified as positive matching.  In fact, the negative point pairs are much more than the positive pairs. If the random sampling strategy is adopted, there will be no positive samples. And this will lead to the crash of the model training. I feel a little confused about the random sampling.
4. In lines 185-190, how to generate dense matching flow through sparse annotations is not well-introduced.
5. In Figure 5, is the inference conducted on the validation or test set? Why there is back-propagation during the inference process? Does NeMF use the output of PatchMatch as pseudo labels to train the model?
6. It seems that NeMF uses the output of PatchMatch as the pseudo label. But there is no ablation study to support it. It will make readers confused about whether the performance gain comes from the model itself or its supervision.
7. It seems that the computational cost of NeMF is relatively high.

---

> ### Author Response · Authors · 2022-08-02
> **Response to reviewer iiVf [1/2]**
>
> We highly appreciate the comments, which our responses are in the following. If any of our responses do not adequately address your concerns, please let us know and we will get back to you as soon as possible.
>
> > **Cost volume construction widely used in other literature**
>
> We wish to highlight that our main contribution does not lie on the "usage" of cost volume but on seamless incorporation of the implicit neural representation (INR) for semantic correspondence task where a coordinate-based neural network allows us to model a continuous matching field. With the learned network, we can implicitly represents a high-dimensional 4D matching field to infer high-precision correspondences at arbitrary scales without any post-processing procedure.
>
> > **Matching ambiguities caused by interpolation.**
>
> We assume that the reviewer is concerned with L168, where we say "use a quadlinear interpolation on C' ...". We agree with the reviewer that there may exist ambiguities when hand-crafted interpolation is used to query to cost feature vector. However, similar to Convolutional occupancy network [E], Plenoxel [F] and NSVF [G], the fully connected layer is the key that alleviates the ambiguities induced from interpolation, because the learning signal sent defined at original resolution rectifies the ambiguously interpolated cost feature, which can help to alleviate the matching ambiguities initially presented.
>
>
> [E] Peng, S., Niemeyer, M., Mescheder, L., Pollefeys, M., & Geiger, A. (2020, August). Convolutional occupancy networks. In European Conference on Computer Vision (pp. 523-540). Springer, Cham.
>
> [F] Fridovich-Keil, S., Yu, A., Tancik, M., Chen, Q., Recht, B., & Kanazawa, A. (2022). Plenoxels: Radiance Fields Without Neural Networks. In Proceedings of the IEEE/CVF Conference on Computer Vision and Pattern Recognition (pp. 5501-5510).
>
> [G] Liu, J., Mantel, C., Schweiger, F., & Forchhammer, S. (2021, November). A Simulation System for Scene Synthesis in Virtual Reality. In International Conference on Virtual Reality and Mixed Reality (pp. 67-84). Springer, Cham.
>
> > **Random sampling during training**
>
> As L180-L182 states, we use the ground-truth matching score for computing the loss. More specifically, given a keypoint p_s at source image, p_t (positive sample) at target image and a fixed number (e.g. 99) of keypoints p_rand (negative samples) randomly sampled, we use cross-entropy loss to maximize the probability of predicting the ground-truth. If we sample too many negative samples, we agree that long-tail problem may be induced, but based on our empirical observations, we did not observe such an issue. We wish to highlight that as the reviewer suggested, we included a discussion regarding the advanced sampling strategy. We would be glad to attempt better sampling strategy in our future work. Thanks for the comment.
>
> > **Dense flow map to sparse keypoints conversion**
>
> Thanks for the comment, we will include one or two sentences to indicate that sparse keypoints are converted to dense flow map following the protocol of CATs [8] (keypoints to dense flow implementation).
>
> > **Is the inference conducted on the validation nor test set?**
>
> During training and validation, we do not use PatchMatch-based sampling and coordinate optimization. These two strategies are designed for the test phase.
>
> > **Why there is back-propagation during the inference process?**
>
> The coordinate optimization requires back-propagation as detailed in Section 4.4. Concretely, because the network is naturally differentiable,  we use a gradient descent to optimize the target coordinate **y** in the direction of decreasing the negative log likelihood of the matching score, with respect to the corresponding source coordinate **x**. A high-level explanation for this is that using the learned network, we want to find the target coordinate that best fits to the given source coordinate to be matched. To this end, we only move around the coordinates in a direction that finds the most corresponding target coordinates.
>
> >**Does NeMF use the output of PatchMatch as pseudo labels to train the model?**
>
> We wish to clarify that the proposed PatchMatch-based sampling is not used as psuedo-labels for training, but it is only used at test-phase. We perform test-time optimization because we optimize the coordinates in a direction that maximizes the correctness of the correspondence using the learned network (L222). As detailed in Section 4.4, although the proposed PatchMatch-based inference strategy could prevent exhaustive searching, this may degrade the performance due to several reasons including insufficient number of iterations and a limited search range. To address this issue, we mitigate the potential erroneous inference by adopting test-time optimization strategy that directly optimize coordinates **y** that maximizes the correctness of the correspondence using the learned the network.

---

> ### Author Response · Authors · 2022-08-02
> **Response to reviewer iiVf [2/2]**
>
> > **Relatively high computational costs**
>
> Although NeMF has higher computational costs than CATs [8],  a simple approximation of memory consumption for conventional approaches would be (h x w)$^2$, which NeMF can avoid this with the proposed approach.  More specifically, we designed the inference strategy in a way that significantly reduces the time taken as shown in Table 3. Also, NeMF can infer correspondences at any arbitrary resolution of images
> with fine-grained details preserved, which is the major contributions of this paper.
>
>
> Note that COTR[31] and DMP [19] are the examples that require highly non-trivial computations and time consumption at inference. Concretely, COTR requires (H x W) / 35 seconds according to the statement "non-optimized prototype implementation queries one point at a time, and achieves 35 correspondences per second on a NVIDIA RTX 3090 GPU.", and DMP requires more than a minute for the complete iterations for optimization. From this, the proposed method has an advantage over them in terms of inference time.
>
> Moreover,  we report memory consumption of our method in comparisons to CATs [8] and CHM [45]. Evaluating with four different resolutions of cost volume, such as 16$^4$, 32$^4$, 64$^4$, 128$^4$, we summarize the memory consumption of both approaches during training and inference as below:
>
>
> | Model | 16$^4$ | 32$^4$  | 64$^4$  | 128$^4$  |
> |---|:---:|:---:|:---:|:---:|
> |  Training CHM | 708 | 1538 | OOM | OOM |
> | Inference CHM | 371 | 433 | OOM  | OOM |
> |  Training CATs | 454 | 3523 | OOM | OOM |
> |  Inference CATs | 188 | 302 | 1882 | OOM  |
> | Training NeMF  |  4205 | 4205 | 4205 | 4205 |
> |  Inference NeMF | 1528 | 1528 | 2443 | 6309 |
>
> As shown in the Table, CATs [8] and CHM [45] suffer from OOM (Out of Memory) as the resolution increases, demonstrating the advantage of the proposed approach implicitly representing matching costs.
>
> >**Biggest difference between NeMF and others**
>
> As explained in L136 to L141, we propose an INR-based learnable framework that implicitly represents high-dimensional matching field to infer correspondences at arbitrary scales. Other works typically perform matching at low-resolution (L33) and requires hand-crafted interpolation techniques. We will highlight our contribution at the end of introduction as the reveiwer a2FL suggested.

---

### Official Review · Reviewer_a2FL · 2022-07-20

**Rating:** 7
**Confidence:** 4
**Soundness:** 3 good
**Presentation:** 3 good
**Contribution:** 3 good

**Summary:**

Deep learning based image matching algorithms rely on creating a 4D tensor representing the cost of matching each pair of pixels. Due to large memory consumption this 4D tensor has to be kept at a coarse scale which affects the quality of matching at a finer scale (i.e., small localization errors are prominent). Inspired from the concept of implicit neural representations, this paper proposes to learn an MLP which can efficiently interpolate in the space of matching costs. To recover the matching during inference, the paper proposes two ways: sampling and gradient based optimization. Experiments show that this approach indeed can match images significantly better at a finer scale than competitive approaches. Overall this idea of implicit neural representation combined with test time optimization can also have applications in other tasks which also compute cost volumes e.g., depth estimation, optical flow etc.

**Questions:**

C. Major questions and suggestions:
1. In equation 6, is $y$ initialized from $F_{pred}$ similar to PatchMatch?
2. In line 272, CATs[8] are said to have a cost tensor of size $16^4$ which at first glance seems to be more than this approach of NMT. However in line 9 of Appendix it is mentioned that NMT needs even more ($16^5$). I would suggest to change the wording in line 272.
3. I did not find details about training time, batch size, compute hardware used for training etc. Also peak memory consumption during training and inference needs to be compared with CATs. I think NMF would need more due to 5D cost representation instead of 4D.
4. The details of cost embedding network 4.2 are fuzzy. In Table 2 please define the components concretely. See Table 1 in [R-MVSNet](https://openaccess.thecvf.com/content_CVPR_2019/supplemental/Yao_Recurrent_MVSNet_for_CVPR_2019_supplemental.pdf) for an example.

D. Minor comments and suggestions (do not need to be addressed in rebuttal):
1. It would be better to have a list of contributions at the end of introduction. This can help the reader in exactly understanding what is novel and what is not in this article.
2. Continuing further on the point B1, have the authors tried making the cost tensor even smaller (e.g. $12^5$) to 'stress-test' the MLP? How worse are the results? It might decrease memory consumption and inference time. I understand if these results cannot be reported within the rebuttal period, but it would be good to have these for final version.
3. Relating to point C4 above, it would be good to know the exact details of architectural changes done on top of CATs possibly in Appendix. Lines 7-9 of Appendix are not detailed enough in my opinion.
3. I assume the concept of neural fields presented in this paper is also interesting for optical flow and depth estimation applications. Thus in broader impact these applications can also be added.
4. The reference of [30] is missing the field of year.


**Limitations:**

Only limitation is more computational burden (compute time and memory) than competitive approaches. This is not a major limitation because the idea itself is novel. Moreover the increase in memory consumption is still linear instead of $O(M^4)$ for $M \times M$-sized images for conventional approaches. Nonetheless these limitations need to be made more clear in the final manuscript.

**Strengths And Weaknesses:**

A. Strengths:
1. The idea of using MLP for recovering fine details from matching costs is novel and possibly useful in other applications as well.
2. The paper is mostly written well and easy to understand.
3. Extensive experiments do show improvement of this approach over competitive approaches on several datasets.
4. Even though this approach incurs slightly more memory burden than best competitive approach (CATs[8]) but it allows to compute the matching costs at any arbitrary resolution foregoing the need for upsampling of the cost tensor.

B. Weakness:
1. The concept of implicit representation was motivated by the need for reducing memory consumption. However, this approach ended-up ultimately using a 5D cost tensor requiring 16x more memory than competitive approach of CATs[8] which uses 4D.
2. Computation: Inference time of this approach is significantly more than competitive approaches. Although the authors do mention this limitation but memory consumption is not reported (for both training and testing).

---

> ### Author Response · Authors · 2022-08-02
> **Response to reviewer a2FL [1/2]**
>
> We thank the reviewer for the thorough comments, which we respond in the following. If any of our responses do not adequately address your concerns, please let us know and we will get back to you as soon as possible.
>
>
> > **NeMF using 5D cost tensor while CATs uses 4D**
>
> In fact, CATs [8] utilizes 5D tensors as it uses multi-level cost maps, resulting in 8 x H_s x W_s x H_t x W_t when SPair-71k is used. This means that the overall computation for cost embedding network would be roughly x2 higher than CATs. Our sincere apologies for ambiguous statements or notations. In addition, we wish to highlight that the proposed method may suffer from relatively larger computation and memory consumption than CATs, but the proposed approach has an advantage when the resolutions of images are high.
>
> > **Memory consumption for training and inference**
>
> Before the response, we wish to correct some errors in L301 and in L310, where we say resolution "512 x 512" and "assuming N = 5", when actual values are "original" and "N = 25" in line with our implementation detail. Sincere apologies for our mistakes and we will correct this.
>
> Following the reviewer's suggestion, we report memory consumption of our method in comparisons to CATs [8] and CHM [45]. Evaluating with four different resolutions of cost volume, such as 16$^4$, 32$^4$, 64$^4$, 128$^4$, we summarize the memory consumption of both approaches during training and inference as below:
>
> | Model | 16$^4$ | 32$^4$  | 64$^4$  | 128$^4$  |
> |---|:---:|:---:|:---:|:---:|
> |  Training CHM | 708 | 1538 | OOM | OOM |
> | Inference CHM | 371 | 433 | OOM  | OOM |
> |  Training CATs | 454 | 3523 | OOM | OOM |
> |  Inference CATs | 188 | 302 | 1882 | OOM  |
> | Training NeMF  |  4205 | 4205 | 4205 | 4205 |
> |  Inference NeMF | 1528 | 1528 | 2443 | 6309 |
>
> As shown in the Table, CATs  and CHM  suffer from OOM (Out of Memory) as the resolution increases, demonstrating the advantage of the proposed approach implicitly representing matching costs. We will include this in the paper. Note that at inference, (16) and (32) share the same memory consumption, and this shows that at very low resolutions, a trivial difference is observed.
>
> Additionally, we wish to emphasize that without affecting the performance, we can reduce computational burden and memory consumption at inference phase by only optimizing the coordinates of interests used for evaluation at coordinate optimization phase and tuning the batch-size of the input coordinates to the PatchMatch-based sampling, which can determine the memory consumption and run-time. Assuming a set of keypoints are for querying is available, we can optimize only the coordinates of the keypoint which we want to find its corresponding keypoint at target image. This way, we can significantly reduce the run-time. The results are shown below:
>
> |  Inference Strategy | Inference Time [s/img] | Memory [MiB] |
> |---|:---:|:---:|
> |  PatchMatch | 1.42 |  6307 |
> | PatchMatch + Optimize all coordinates | 2.21 | 6309 |
> |  PatchMatch + Optimize only keypoints | 1.65  | 6308 |
>
> For this experiment, we assumed NeMF is representing cost volume of size 128$^4$ to show the memory comparison that will be presented below this paragraph.  From the table above, we observe negligible memory change, but significant reduction in run time when only the keypoints of interest are optimized. With this strategy, we can benefit from the reduced run-time, and by reducing the batch-size of the input coordinates to the PatchMatch-based sampling we can also control the memory consumption. For this experiment setting, we use the default coordinate batch size setting, which is 100,000 and cost volume size of 128$^4$:
>
> | Batch Size | Inference Time [s/img] | Memory [MiB] |
> |---|:---:|:---:|
> |  100000 | 1.65  |  6308 |
> | 50000 | 2.27 | 4301 |
> |  25000 | 4.43  | 2730 |
> |  10000 | 9.18 |  1789 |
>
> This table shows that tuning the batch-size can reduce the memory consumption by sacrificing run-time, meaning that users can choose to infer with high/low memory and fast/slow run-time.
>
> > **$y$ initialized similar to PatchMatch?**
>
> As stated in L204, we utilize an average pooled cost feature volume for the initialization step, which is different from PatchMatch.
>
> > **L272 wording**
>
> As mentioned to the comment above, we actually utilize the 4D cost tensor obtained by average pooling the 5D cost feature. We will make this clearer in the paper. Thanks for the comment.
>
> > **Missing details**
>
> We train our networks with Geforce RTX 3090 with batch-size 20 for 20 epochs, which takes around 16 hours. For the implementation details, we build our cost embedding network upon the architecture of CATs. Specifically, notable modifications include convolutions prior to CATs architecture as justified in experiments of Table 2 and excluded appearance affinity modelling, which was adopted in CATs (concatenation of projected feature maps to cost volume). We will clarify this in the implementation detail and Section 4.2.

---

> > ### Comment · Reviewer_a2FL · 2022-08-04
> > **Post-rebuttal comment**
> >
> > Thanks to the authors for detailed answers. My questions are adequately addressed and the authors do acknowledge limitations (and benefits): _"we wish to highlight that the proposed method may suffer from relatively larger computation and memory consumption than CATs, but the proposed approach has an advantage when the resolutions of images are high."_. Please discuss this statement in the camera ready version as well.
> >
> > I am continuing to recommend acceptance.

---

> ### Author Response · Authors · 2022-08-02
> **Response to reviewer a2FL [2/2]**
>
> > **Concrete components to be held in Table 2**
>
> We will include pseudo-code or a table as in R-MVSNet [C] for Table 2 in supplementary material as the reviewer kindly suggested. Note that for the concerete components in Table 2, NeMF was applied to 4D volume which was obtained with the following cases;
>
> (I). A raw correlation map constructed from a pair of feature maps,
>
> (II). 4D volume applying the center-pivot convolutions (HSNet [D]) to the raw correlation map,
>
> (III). 4D volume applying CATs (without appearance affinity modeling) to the raw correlation map
>
> (IV). Combination of (II) and (III), i.e., 4D volume applying the center-pivot convolutions and CATs (without appearance affinity modeling) sequentially.
>
> [C] Yao, Y., Luo, Z., Li, S., Shen, T., Fang, T., & Quan, L. (2019). Recurrent mvsnet for high-resolution multi-view stereo depth inference. In Proceedings of the IEEE/CVF conference on computer vision and pattern recognition (pp. 5525-5534).
> Note that the coordinate optimization was commonly applied to four cases.
>
> [D] Min, J., Kang, D., & Cho, M. (2021). Hypercorrelation squeeze for few-shot segmentation. In Proceedings of the IEEE/CVF International Conference on Computer Vision (pp. 6941-6952).
>
> > **Stress test**
>
> We would like to thank the reviewer for suggesting an interesting direction. Although we were unable to obtain the complete quantitative result on SPair-71k for the stress test, we measured the memory consumption and the summary is shown below. Note that we conducted this experiment with 128$^4$ resolution, and 100,000 batch-size for coordinates:
>
>
> | Phase | Resolution | Memory [MiB] |
> |---|:---:|:---:|
> |  Training | $12^4$ |  3766 |
> | Training | $16^4$ | 4205 |
> |  Inference | $12^4$  | 4048 |
> |  Inference | $16^4$ | 6308 |
>
> We observed that by changing the resolution of the cost volume, the memory is greatly reduced both in training and inference. Once the training finishes, we will compare with the best results to see if this stress test substantiates its effectiveness.
>
> > **D4 and D5**
>
> Thanks for the valuable suggestion and pointing out the absence of field of year. We can add in the broader impact that the proposed approach that implicitly represents high dimensional matching fields has a potential for a general use to those works that employ high resolution cost volumes with fine-grained information preserved, e.g., semantic segmentation and optical flow.

---

### Official Review · Reviewer_JRSW · 2022-07-26

**Rating:** 5
**Confidence:** 4
**Soundness:** 3 good
**Presentation:** 3 good
**Contribution:** 2 fair

**Summary:**

This paper introduces neural matching fields into semantic correspondence. To the best my knowledge, this approach should be the first method to do the task using implicit neural representation. There are two problems: the computation for 4D matching field and the inference efficiency. Authors provide effect method to address the two problems.

**Questions:**

I only have one concern. Traditional Implicit Neural Representation method such as LIIF and NeRF records images into the weights of neural network. One neural network represents one image or one scene. Does NeMF take a neural network to represent a semantic correspondence or a matching cost.   If so, how much time will your method cost to train a network? If not so, what is the difference between your method and other semantic correspondence methods.

**Limitations:**

According to my understand, NeMF takes a network to represent a matching cost. In practice, people need a method to compute different matching cost for different image pairs. How does NeMF to deal with this situation.

**Strengths And Weaknesses:**

This paper employs implicit neural representation to do semantic correspondence. This should be the major contribution. According to the statement of authors, I can follow the idea easily and this idea should work.

The disadvantage of this work is the experiments. There are too many quantitative comparisons. According to the data, the performance of this method seems OK. However, authors should provide more visual experiments to convince readers.

---

> ### Author Response · Authors · 2022-08-02
> **Response to reviewer JRSW**
>
> We thank the reviewer for the positive assessment of our paper. If any of our responses below do not adequately address your concerns, please let us know and we will get back to you as soon as possible.
>
> > **Too many quantitative comparisons and more visual experiments**
>
> We thank the reviewer for the positive assessment of our paper. We agree that it is a good idea to provide more visual experimental results. Nevertheless, we wish to mention that Fig.2 and Fig.6 visualize matching fields and flow maps to help the understanding of readers. Furthermore, qualitative results on each benchmark are given in supplementary material. However, following the reviewer's suggestion, we will reduce the number of compared methods in Table 1, and replace with other visual experiments, including warped source images with evolving iterations as in DMP [19].
>
> > **Does NeMF take a NN to represent a semantic correspondence or a matching cost?**
>
> Thanks for the question. First, we wish to clarify that the original NeRF adopts an *unconditional* implicit neural representation (INR) encode a single scene or an image. On the other hand, the literature that utilize *conditional* INR, e.g., PixelNeRF [77], IBRNet [A], and Mvsnerf [B] overcome this shortcoming by conditioning a NeRF on an input image to learn a scene prior.
>
> Regarding the question, NeMF represents matching cost based on the *conditional* INR framework, where a matching prior can be learned by conditioning the network on the matching cost features.
>
> [A] Wang, Q., Wang, Z., Genova, K., Srinivasan, P. P., Zhou, H., Barron, J. T., ... & Funkhouser, T. (2021). Ibrnet: Learning multi-view image-based rendering. In Proceedings of the IEEE/CVF Conference on Computer Vision and Pattern Recognition (pp. 4690-4699).
>
> [B] Chen, A., Xu, Z., Zhao, F., Zhang, X., Xiang, F., Yu, J., & Su, H. (2021). Mvsnerf: Fast generalizable radiance field reconstruction from multi-view stereo. In Proceedings of the IEEE/CVF International Conference on Computer Vision (pp. 14124-14133).
>
> > **How much time does the proposed method take for training?**
>
> Equipped with 4 Geforce RTX-3090 GPUs, a single epoch with batch size of 20 on SPair-71k dataset takes about 50 minutes, resulting about 16 hours to complete the whole training. We will include this in the implementation detail.
>
> >**Different matching cost for different image pairs. How does NeMF cope with this?**
>
> Our method can adaptively infer an intrinsic matching prior no matter what image pair is given as input by conditioning NeMF on the matching cost feature computed from the input images. This is in contrast to the previous approaches based on unconditional INR where image-specific prior can disrupt the generalization capacity.

---

### Author Response · Authors · 2022-08-03
**General Response**



We thank all the reviewers for their comments, and are glad that the reviewers found our work "novel and useful in other applications” (a2FL, vfUA), “easy to follow and understand" (JRSW, iiVf), and “supported by extensive experiments” (a2FL, iiVf). In this rebuttal, we mainly addressed following points:

1. Detailed explanation of NeMF as a conditional INR based framework (JRSW).
2. Memory consumption comparison to show that the proposed method takes an advantageous approach to represent high-resolution matching cost (a2FL, iiVf).
3. Details of training and cost embedding network (a2FL, vFUA) and clarified the motivations.
4. Contributions and methodology (iiVf).
5. Additional minor comments.

---

### Author Response · Authors · 2022-08-09
**Author-Reviewer discussion**

Dear reviewers,

Since the rebuttal discussion is about to end soon, if there is any other concern that we did not adequately address or is not resolved, please let us know, and we will come back to you as soon as possible if we can.

Thank you and best regards,

The authors of Paper 2300.

---

### Meta-Review · Area_Chair_Wr6P · 2022-08-23

**Recommendation:** Accept
**Confidence:** Less certain

**Metareview:**

The paper concerns itself with computing high resolution matchings. The authors propose to use represent matchings as maxima of neural "matching" fields, which is a novel and interesting theoretical contribution that allows to obtain high resolution matchings with fixed representation size of the neural field. The matchings are extracted from the neural field via coordinate optimization. State of the art performance is attained on a variety of semantic correspondence benchmarks. Reviewers also acknowledge that the paper is well written and easy to follow. On the downside are larger computational costs. Also newer versions of CATS, i.e. CATS++ (which is concurrent work), outperform the presented paper. Overall, the interesting theoretical contribution that might be useful in other domains and strong empirical performance make the paper a good fit for NeurIPS.
In a final version the reviewer recommendations must be taken into account.

**Award:**

No

---

### Decision · Program_Chairs · 2022-09-14

Accept